# Modifiable Maternal Factors and Their Relationship to Postpartum Depression

**DOI:** 10.3390/ijerph191912393

**Published:** 2022-09-29

**Authors:** Kathryn Howard, Jill M. Maples, Rachel A. Tinius

**Affiliations:** 1Biology Department, Western Kentucky University, Bowling Green, KY 42101, USA; 2The Department of Obstetrics and Gynecology, University of Tennessee Graduate School of Medicine, Knoxville, TN 37996, USA; 3School of Kinesiology, Recreation, and Sport, Western Kentucky University, Bowling Green, KY 42101, USA

**Keywords:** postpartum, depression, fatigue, sleep, physical activity, breastfeeding

## Abstract

The purpose of the study was to examine how modifiable maternal factors (body mass index (BMI), household income, fatigue, sleep, breastfeeding status, diet, and physical activity) relate to postpartum depression (PPD) at 6 and 12 months postpartum. Participants (n = 26) participated in two study visits (6 and 12 months postpartum) where vitals, weight, body composition (skinfold anthropometrics), and physical activity levels (Actigraph GTX9 accelerometer) were assessed. Validated instruments (BRUMS-32, Subjective Exercise Experience Scale, Pittsburg Sleep Quality index, NIH breastfeeding survey, NIH Dietary History Questionnaire, and Edinburg Postnatal Depression Scale) assessed lifestyle and demographic factors of interest. PPD at six months was correlated to PPD at 12 months (r = 0.926, *p* < 0.001). At six months postpartum, PPD was positively correlated to BMI (r = 0.473, *p* = 0.020) and fatigue (r = 0.701, *p* < 0.001), and negatively correlated to household income (r = −0.442, *p* = 0.035). Mothers who were breastfeeding had lower PPD scores (breastfeeding 3.9 ± 3.5 vs. not breastfeeding 7.6 ± 4.8, *p* = 0.048). At 12 months, PPD was positively correlated to sleep scores (where a higher score indicates poorer sleep quality) (r = 0.752, *p* < 0.001) and fatigue (r = 0.680, *p* = 0.004). When analyzed collectively via regression analyses, household income and fatigue appeared to be the strongest predictors of PPD at six months postpartum.

## 1. Introduction

After birth, the focus of attention shifts from the health of the mother to that of the baby [1]. The perceived lack of importance placed on the mother’s health after the birth of the child can lead to negative long-term health outcomes for the mother and child, including postpartum depression (PPD) [2]. PPD is a serious issue affecting one in six new mothers [2]. Symptoms of this condition include irritability, anhedonia, anxiety, persistent discouragement, guilt, and other symptoms [3]. These symptoms begin to present themselves around four to six weeks after delivery [3]. PPD can lead to problems with mother–infant bonding and may also contribute to mothers being less likely to engage in infant safety practices and attend health care visits [2]. The symptoms of depression can persist for months or even years after the birth of the child, increasing the risk of maternal suicide [3].

Previous research on PPD has shown that there is no single cause [4]. There have been several etiologic risk factors identified including genetic variants, hormonal fluctuations, history of depression, personality, socioeconomic status, adolescent pregnancy, and unintended pregnancy [4], many of which cannot be modified or changed. The modifiable maternal factors assessed in the present study have been previously analyzed but have not been assessed together. The present study assessed many factors known to effect mental health following pregnancy together, prospectively, to determine which factors could be potentially modified to improve health outcomes.

Understanding the specific factors that can aid postpartum recovery and improve outcomes is important and not well studied. The present study focuses specifically on risk factors that can be potentially modified to improve health outcomes for women. Therefore, the purpose of the present study is to determine the correlations between PPD and modifiable maternal factors including body mass index (BMI), household income, fatigue, sleep, breastfeeding status, diet, and physical activity. The results of the study will serve as a crucial step towards identifying factors to target and improve the mental (and physical) health of new mothers. 

## 2. Materials and Methods

Data were collected as part of a larger study among postpartum people in southcentral Kentucky [5]. Recruitment for the present study took place between 32 and 34 weeks of gestation. All women participated in a study during their pregnancy [6] and were then recontacted to participate in the postpartum follow-up study visits. The postpartum follow-up study involved attending two additional visits, one at 4–6 months and the other at 12–13 months postpartum. 

To be eligible to participate in the study, the women had to meet inclusion and exclusion criteria from the initial study (previously reported) as well as report no new medical conditions for themselves or the baby that could influence metabolic health [6]. The procedures for the present study were approved by the Institutional Review Board of Western Kentucky University (IRB 17-412). Before each visit, the participants were provided with a written informed consent form, which was verbally explained prior to obtaining a signature while present in the lab. 

Study visit procedures were the same at both time points (6 and 12 months). Baseline vitals, including resting heart rate, blood pressure, and anthropometrics (height, weight, body fat percentage) were obtained. Physical activity was assessed via accelerometry. The participants wore an Actigraph GTX9 Link accelerometer (Actgraph LLC, Pensacola, FL, USA) on their non-dominant wrist for seven consecutive days. The participants also completed (pencil and paper) a series of surveys and questionnaires while physically present in the lab during the study visit. These included questions related to sleep, breastfeeding status, mental health, and diet.

The Pittsburgh Sleep Quality Index (PSQI) was used to measure the quality and patterns of sleep. The assessment differentiates “poor” from “good” sleep by measuring the following areas: subjective sleep quality, sleep latency, sleep duration, habitual sleep efficiency, sleep disturbances, use of sleeping medications, and daytime dysfunction over the last month. The PSQI has been validated for use on women who are pregnant [7]. Researchers analyzed the responses and assigned each participant with a score to indicate if they were sleep deprived or getting enough sleep. A higher score indicates poorer overall sleep quality. 

Participants completed an extensive NIH breastfeeding survey regarding amounts of breastmilk vs. formula, timing of feedings, and other foods the infant is eating. For the present study, we only considered if the participants were breastfeeding, formula feeding, or a combination of both. 

The Edinburg Postnatal Depression Scale (EPDS) is a 10-item self-report scale that is used to screen for PPD. Scores range from 0 (lowest depression score) to 30 (most depressed score). Possible depression exists if a score of 10 or higher is received. The EPDS has been widely used since its development and has shown satisfactory sensitivity and specificity [8]. 

The Brunel Mood Scale (BRUMS-32) measures eight identifiable affective states by using a 32-item self-report inventory where respondents rate a list of adjectives on a 5-point Likert scale. The ratings range from 0 (not at all) to 4 (extremely) and are based on subjective feelings [9]. The scores from the eight mood states (anger, tension, depression, vigor, fatigue, happy, confusion, and calmness) are used to create sub scores for positive well-being (PWB), physiological distress (PD), and fatigue (FAT). BRUMS has been validated for healthy physically active populations by consistent results [10].

The Subjective Exercise Experience Scale (SEES) is a 12-item scale used to assess subjective feelings at one point in time, including positive well-being, physiological distress, and fatigue [9,11]. The SEES measures can also be used to assess psychological responses to exercise (data not included in present study). The SEES has been validated for the use of assessing mood [11], particularly fatigue, which is how it was used in the present study. 

The NIH Dietary History Questionnaire II (NIH DHQII) (with portion sizes) was used to determine dietary information on study participants at 6 and 12 months postpartum. The NIH DHQII is a 135-item questionnaire that determines macro and micronutrient contents of the diet over the past month [12]. 

All data analyses were conducted using IBM SPSS Statistics, Version 28 (Armonk, New York, NY, USA). Shapiro–Wilk tests were conducted to determine normality of the data. For continuous variables that were not normally distributed, log transformations were utilized to normalize distributions. Means and standard deviations were calculated for continuous variables and, for categorical variables, counts and percentages were tabulated. To examine relationships between PPD and other variables, Pearson Product Moment Correlation Coefficients were used, and Spearman Rank Order Correlation Coefficients were used for non-normally distributed variables. When appropriate, PPD scores were compared between groups using independent t-tests (e.g., comparing PPD scores between breastfeeding and non-breastfeeding women). Simple and multiple linear regression models determined independent factors affecting PPD at six months and significant predictors. Predictors were included in the regression model based on a *p*-value threshold of *p* < 0.10. Regression analysis was not completed for PPD at the 12-month time point due to low sample size (N = 17). 

## 3. Results

### 3.1. Demographic Characteristics

Descriptive statistics for demographic variables for participants at all time points can be found in Table 1. Overall, 63% of participants started pregnancy at a normal/healthy body mass index, 62% were multiparious, were an average of 32 years old, were all white, and had an income level well above the US median. 

### 3.2. EPDS

At six months postpartum, 4 of the 24 participants who took the EPDS survey scored 10 or higher indicating possible depression. At 12 months postpartum, 17 participants completed the EPDS. Three of the seventeen participants who took the survey scored 10 or higher, indicating possible depression. PPD at six months was strongly correlated to PPD at 12 months, (r = 0.926, *p* < 0.001) (Figure 1). 

### 3.3. BMI

Pre-pregnancy BMI and PPD at six months were positively correlated, (r = 0.468, *p* = 0.021), and Figure 2 shows average EPDS scores when grouped by weight status, and when combined women who are overweight and obese had higher EPDS scores than women with a normal pre-pregnancy BMI (*p* = 0.047). Pre-pregnancy BMI and BMI at both 6 and 12 months were highly correlated (six months: r = 0.921, *p* < 0.001 and 12 months: r = 0.867, *p* < 0.001). 

### 3.4. Household Income

Household income and PPD scores were not independently correlated at six months, (r = −0.391, *p* = 0.065) or at 12 months (r = −0.154, *p* = 0.556). However, when adjusted for other factors in a regression model, household income does appear to predict PPD (*p* = 0.037). 

### 3.5. Fatigue

Fatigue data were collected via two different methods (BRUMS and SEES). Fatigue scores for both surveys were correlated to each other at each time point (six months: r = 0.731, *p* < 0.001; 12 months: r = 0.889, *p* < 0.001), substantiating the quality of these data collections (see Appendix A).

The more fatigued the women were, the more likely they were to experience PPD. Fatigue scores as assessed by the SEES were significantly correlated to PPD scores at both time points (SEES: six months: r = 0.642, *p* < 0.001; 12 months: r = 0.679, *p* = 0.003) (Figure 3). Relationships between fatigue from the BRUMS and PPD were also highly correlated (six months: r = 0.701, *p* < 0.001; 12 months: r = 0.680, *p* = 0.004).

Pre-pregnancy BMI and fatigue scores (per the SEES) were positively correlated at both six months (r = 0.410, *p* = 0.047) and 12 months (r = 0.636, *p* = 0.006). Postpartum BMI and fatigue scores were positively correlated at both timepoints: 6 months (BRUMS, r = 0.500, *p* = 0.015) and 12 months (SEES, r = 0.509, *p* = 0.037). The relationships between fatigue and PPD did not change when controlling for BMI or breastfeeding status at either timepoint. 

### 3.6. Sleep

At 6 months postpartum, PPD scores and sleep scores were not correlated (r = 0.076, *p* = 0.731); however, they were positively correlated at 12 months postpartum (r = 0.752, *p* < 0.001) (Figure 4). Moms who reported better sleep had lower PPD scores (a higher component score on the PSQI indicates lower sleep quality). When grouped by sleep category, those with better sleep scores had lower PPD scores (*p* < 0.001).

Fatigue (using both scales, BRUMS and SEES Fatigue Subscale) and quality of sleep (PSQI) were not correlated at 6 months postpartum (BRUMS: r = 0.326, *p* = 0.139, SEES: r = 0.139, *p* = 0.527), but were at 12 months postpartum (BRUMS: r = 0.624, *p* = 0.010, SEES: r = 0.538, *p* = 0.026). 

### 3.7. Breastfeeding

At six months postpartum, mothers who reported breastfeeding had lower PPD scores (breastfeeding:3.9 ± 3.5 vs. not breastfeeding:7.6 ± 4.8, *p* = 0.048) (Figure 5). Breastfeeding status was not related to PPD when controlling for pre-pregnancy BMI.

### 3.8. Physical Activity

At six months, PPD was not related to physical activity. Activity was measured in three categories and correlations between each category and PPD scores were determined: sedentary (−0.018, *p* = 0.937), light (r = −0.018, *p* = 0.943), and moderate (r = 0.098, *p* = 0.665), and these values were similar (not significant) at 12 months: sedentary (0.270, *p* = 0.331), light (r = −0.256, *p* = 0.357), and moderate (r = −0.263, *p* = 0.345).

### 3.9. Dietary Intake

Previous work suggests a relationship between certain dietary components (total calories, vitamin B6, zinc, and selenium) and PPD [13]. At both time points, correlations were not found between total energy, macronutrient, or micronutrient intake and PPD (Table 2).

### 3.10. Simple and Multiple Regression

A multiple linear regression was calculated and a significant regression equation predicting PPD at six months postpartum was found (F(4,17) = 6.189, *p* = 0.003), with an adjusted R-squared of 0.497. When adjusting for pre-pregnancy BMI, fatigue, and infant feeding status, income was a significant predictor of PPD at six months postpartum (*p* = 0.037). Only fatigue score remained a significant predictor (*p* = 0.007) of PPD at six months postpartum when adjusting for pre-pregnancy BMI and infant feeding status (Table 3). 

## 4. Discussion

The modifiable maternal factors BMI, household income, fatigue, breastfeeding status, and sleep can each impact PPD. Figure 6 summarizes the findings of the present study. PPD was positively correlated to pre-pregnancy BMI and fatigue, while PPD was negatively correlated to household income and sleep quality. Additionally, mothers who breastfeed have lower PPD scores. Many of these findings are consistent with the previous literature [14,15,16,17]. However, when analyzed together in a regression model, it is clear that the aforementioned variables have complex interactions with each other and PPD. Our study is novel in that data on all these different factors were collected prospectively among one study cohort during the first year following birth. Most studies exploring PPD and factors that contribute to it are retrospective and focus on one factor (e.g., sleep). Gaining a better understanding of how these modifiable maternal factors are related, and may interact to influence PPD, can help new mothers prepare for the mental health challenges experienced after giving birth.

Figure 6 summarizes the study findings and solidifies the complex relationship between many factors that influence postpartum depression. The dotted line represents the notion that breastfeeding status does seem to play a role in postpartum depression; however, this relationship is modulated by maternal pre-pregnancy BMI. 

New mothers need resources to aid them during this new and challenging experience. It is well established that, after the baby is born, the focus of medical attention shifts to that of the infant [1], and oftentimes maternal health is disregarded. This is represented by the fact that during pregnancy, women attend between 10 and 15 prenatal visits while after birth, the previous standard is one six-week follow-up appointment for postpartum care. The American College of Obstetricians and Gynecologists has recognized the lack of postpartum care and recently recommended a follow-up visit at three weeks and a comprehensive visit at 12 weeks [18]; however, most obstetric practices have not shifted their practices in accordance with these guidelines. Regardless of the number of allowable visits as part of routine care, up to 40% of women do not attend either of the recommended postpartum visits [18].

The American College of Obstetricians and Gynecologists recommends that healthcare providers screen mothers at least once for depression and anxiety symptoms during the perinatal period using a standardized and validated tool and that they complete a full assessment of mood and emotional well-being during the comprehensive postpartum visit [19]. Several risk factors for PPD have been previously identified, but the true cause is not yet to be understood and likely differs per individual [20]. Previous research suggests that the cause of PPD is multifactorial, likely including a combination of mental, socioeconomic, physical, and psychosocial factors [20]. The present study is consistent with this idea and reports on multiple modifiable maternal factors and their significant relationships to PPD. 

Pre-pregnancy BMI and PPD were positively correlated at 6 and 12 months. Depression scores were higher among women who were overweight and obese compared to women with normal weight status, and BMI modulated the relationship between other variables, specifically breastfeeding status, and PPD. 

Previous work has described a relationship between weight status and PPD. Silverman et al. reported extremes in BMI (in either direction) are associated with an increased risk of PPD [21]. A 2018 study found that women who have a history of depression are at a higher risk of having a low BMI [21]. Being overweight may increase the risk for PPD to a greater degree for women who do not have a history of depression [21]. For the present study, pre-pregnancy mental health data were not collected due to the inherent challenges of collecting data on a woman before she is pregnant (i.e., it is not possible to know who will become pregnant). This limits correlation of the presence of depression pre-pregnancy with BMI and PPD. The present study found that women who were overweight or obese had higher PPD scores, which is consistent with previous studies. Women who are obese specifically are significantly more likely to report emotional or traumatic stressors during pregnancy than women of normal weight [22]. In 2022, a meta-analysis of observational studies determined that excessive gestational weight gain was significantly associated with PPD [23]. 

One potential explanation for the finding of the present study is that women who fall into the overweight or obese category could be more susceptible to body image dissatisfaction resulting in higher PPD scores. It is well-established that women with obesity are more likely to experience body weight dissatisfaction [24,25] and that body weight dissatisfaction is linked to postpartum depression [26,27]. 

Women with higher incomes had lower PPD scores. Previous studies show that having a low household income increases the chances of PPD development [14]. The mental health condition of a new mother can be impacted due to lack of resources, access, and support [28]. One weakness of the present study is that many of the women represented were middle to upper class, so the implications of low socioeconomic status on the mental health of mothers could not be determined. While each of the maternal factors analyzed in the present study are potentially modifiable, it should be acknowledged that some can be modified more easily than others. Income, specifically, is a factor that women in certain life circumstances may not have the means to change. It is certainly possible that some mothers may experience a drop in pay during the postpartum period if unable to work for a period of time with a job that does not have maternity leave policies in place. Some women may also opt for extended unpaid leave which may impact income. 

Fatigue also had a positive correlation with PPD scores in the present study, which is consistent with previous work. A meta-analysis concluded there was a strong correlation between fatigue and depressive symptoms during the first two years after birth in parents [15]. Fatigue is one of the presenting symptoms of depression and is part of the diagnostic criteria for depressive disorders [29]. The question of whether fatigue causes PPD or if PPD causes fatigue has yet to be answered, and it is likely a vicious cycle, but the findings presented in the present study support the existing idea that PPD and fatigue are correlated.

In the present study, sleep was negatively correlated with PPD at 12 months. Other studies have supported the finding that poor sleep quality is related to increased depression and anxiety [16]. One possible explanation for this finding could be the continued sleep disturbances with decreasing support from those around the mother. It is well established that a major challenge for new mothers is balancing all the competing demands in her life [30]. It appears that maternal sleep postpartum is fragmented; this fragmentation often persists for months after having a baby [31]. At six months, new mothers may be receiving more support and understanding from those around her. By 12 months, mothers may be accumulating months of fragmented sleep and sleep debt [32], with the possibility of less support and understanding from others. The continued disrupted sleep but lack of support and understanding from those around the mother could contribute to the increased PPD scores. Additionally, work has found that, at seven months postpartum, minimal improvement of sleep problems was associated with depressive symptoms [33]. It is possible that the longer the sleep problems linger, the more likely they are to contribute to feelings of PPD. 

In the present study, at six months postpartum, mothers who breastfed had lower PPD scores, which is consistent with previous work. A systematic review of 48 studies on breastfeeding and depression determined that breastfeeding and PPD are highly correlated [17]. Additionally, depression during pregnancy predicts shorter breastfeeding duration [17], thus creating a cyclic relationship between postpartum depression and unsuccessful experiences with breastfeeding. One study revealed that women who were unable to meet their prenatal breastfeeding expectations had higher score on the EPDS [34]. Most women in the United States intend and initiate breastfeeding, but many do not meet their breastfeeding expectations [34]. Being unable to meet breastfeeding expectations set for oneself can be a cause of the relationship between a lack of breastfeeding and PPD [34]. These finding can be applied by informing expecting mothers of realistic expectations regarding breastfeeding [34]. There is evidence of a bidirectional association between maternal mental health status and breastfeeding [35]. Interventions offering individualized support (from professionals and peers in a variety of settings) have been successful at improving both mental health and breastfeeding outcomes [35]. 

Physical activity and PPD were not related in the present study, which was an unexpected finding. A meta-analysis of 12 studies on the effect of physical activity during pregnancy and the postpartum period on PPD found that physical exercise during pregnancy and the postpartum period can be used to achieve improved physiological well-being and reduce PPD scores [36]. It is possible that the women in the present study were not doing high enough intensity exercise to impact mood [9] as moderate (and higher) intensity exercise levels were low among the entire sample. Many of the studies analyzed in the aforementioned meta-analysis were interventions with structured exercise regimens. The women in the current study were not required to do any sort of structured activity; activity levels were measured in real time in normal daily activities. However, we did use high-quality objective tools (Actigraph Accelerometers); some previous work used subjective assessments, which may be less accurate and subject to bias. 

Dietary intake and PPD were not related in the present study. Previous work found that reduced intake of the certain micronutrients (zinc, selenium, and vitamin B6) was linked to the prevalence of PPD [13]. Findings from the present study suggest that factors other than dietary intake may have a greater impact on PPD. 

The main limitation to the current study is the small sample size; however, the cohort was established by recruiting from a pregnancy study previous conducted [6], so the number of women eligible to participate was small (N = 64). All 64 women had new babies and were adjusting to parenthood, as well as going back to work, making it a challenge to have them return for two study visits during the postpartum period. Of note, the sample size was particularly small at 12 months, which is why a regression analysis was not conducted for this time point. Of note, many relationships were detected between variables of interest and PPD at 6 months but not 12 months, which is likely due to the reduced sample size at 12 months. However, because there is a strong relationship between PPD at 6 and 12 months, it is reasonable to assume the factors contributing to PPD at 6 months may continue to contribute to mental health status at 12 months. 

Another limitation is the lack of mental health data from prior to delivery and even prior to pregnancy. Being able to compare the PPD data to the mental health conditions of the women before delivery/pregnancy would provide some novel insight and enhanced understanding of the role of all the studied factors for developing PPD. Women with a history of depression are 20 times more likely to suffer from PPD than women without [37]. Future studies should also consider recruiting women with gestational diabetes as women with gestational diabetes are at higher risk for PPD [37]. The present pregnancy study excluded women with gestational diabetes. 

An additional limitation of the study is the lack of racial diversity in the participant population. All the participants were white. According to the 2021 U.S. census, 72.5% of the population in Bowling Green, Kentucky is white [38]. Thus, the study sample is not a direct representation of the racial breakdown of pregnant women in Bowling Green, Kentucky. Of the 64 women recruited for the original study, 62 were white, 1 was Hispanic, and 1 was African American. Previous research has shown that many sociodemographic factors negatively impact racial minorities [39]. The negative impact of these factors could affect PPD scores. African American and Hispanic mothers have a higher risk of reporting early PPD symptoms in comparison to white counterparts [40]. It should be noted that the findings of this study are only applicable to women who are white and middle or upper-class. Future studies should make an effort to include underrepresented groups who are more likely to experience poor pregnancy outcomes including PPD.

The present study has notable strengths. These include reporting on multiple modifiable maternal factors and the relationship to PPD. Each of these factors have been previously studied but not analyzed together in one study (both independently and as part of regression analyses). The data were also captured prospectively. The physical activity data were collected quantitatively using state-of-the-art technology. Additionally, one of the key variables studied, fatigue, was collected with two different instruments (SEES and BRUMS) and not only did they correlate well to each other (validating our findings), but they also had the same relationships with all other measured variables, which reinforces these relationships and suggests they were not found by chance. Another strength is that even survey data were collected while in the laboratory with participants. In the era of modern technology, so much data are collected electronically. While convenient, there is no way to ensure the participant is carefully reading questions and answering them honestly. In addition, electronic survey collection data do not allow for the ability for participants to ask questions or seek clarification, which the present study did allow for. 

## 5. Conclusions

Body mass index, household income, fatigue, sleep, and breastfeeding status all contribute to PPD in unique and interactive ways. These factors should be carefully considered and modified, when possible, for new mothers in order to improve mental and physical health. Because PPD creates an environment that is detrimental to the health of the mother and the health of the baby [41], it critical to develop intervention strategies to target factors most closely linked to the emergence of PPD and PPD symptoms. Early detection and treatment of modifiable maternal factors that increase the likelihood of PPD must improve to prevent harmful consequences. 

## Figures and Tables

**Figure 1 ijerph-19-12393-f001:**
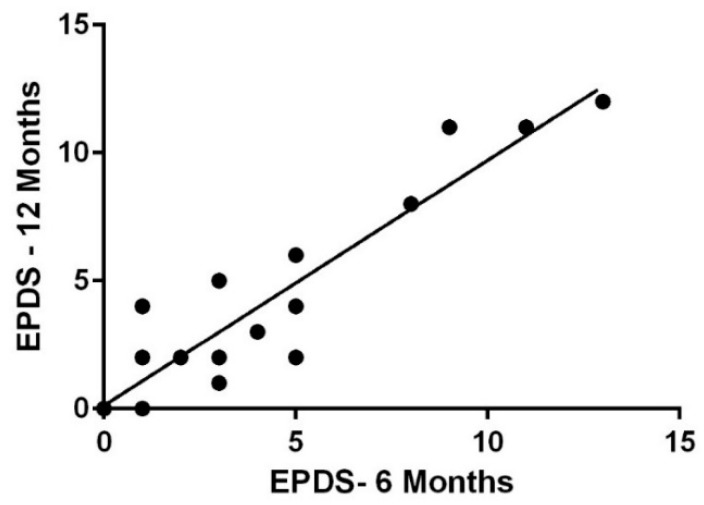
EPDS Scores at Six Months Postpartum compared to EPDS Scores at 12 Months Postpartum.

**Figure 2 ijerph-19-12393-f002:**
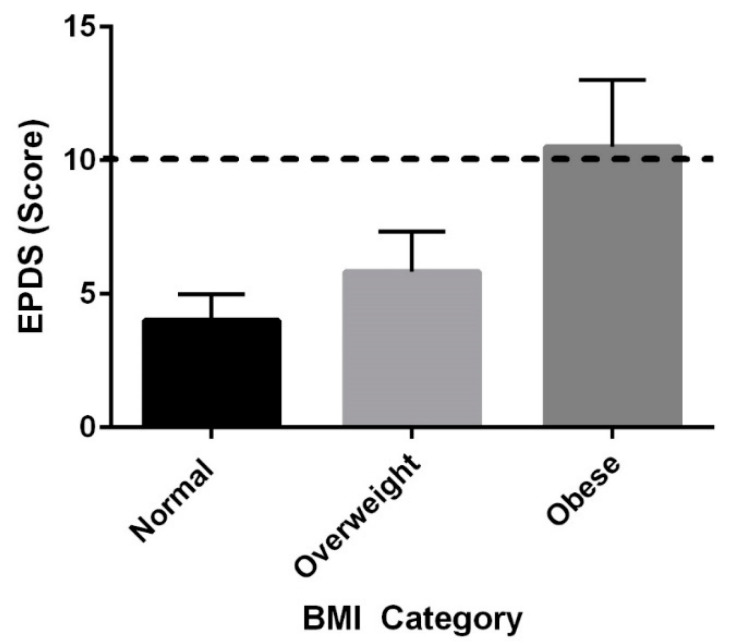
Mean ± SE EPDS Scores at Six Months Postpartum between pre-pregnancy BMI categories; ------- signifies indicator for possible PPD.

**Figure 3 ijerph-19-12393-f003:**
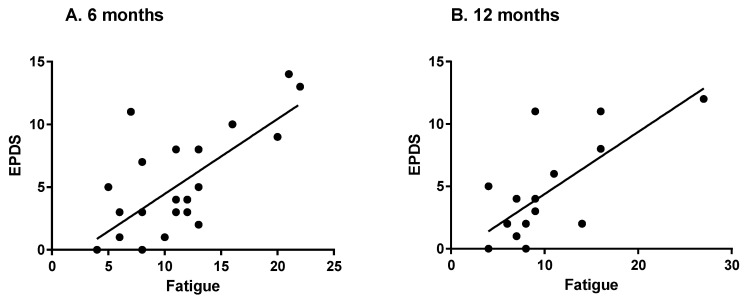
EPDS scores and fatigue at 6 months postpartum (**A**) and 12 months postpartum (**B**).

**Figure 4 ijerph-19-12393-f004:**
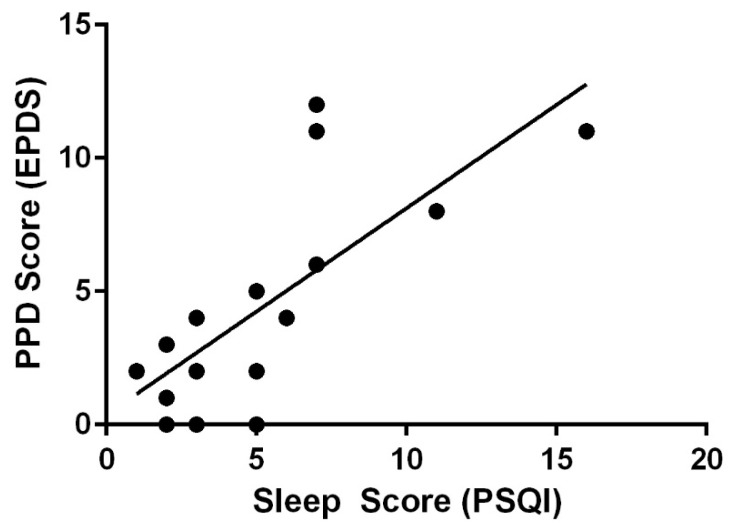
Sleep and PPD Scores at 12 months.

**Figure 5 ijerph-19-12393-f005:**
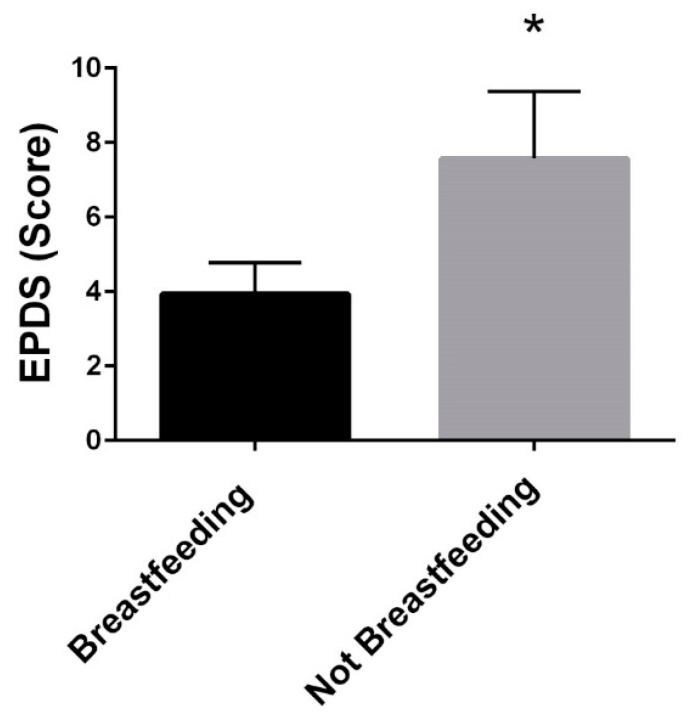
EPDS Scores at Six Months Postpartum between mothers who breastfeed and those who do not. * *p* < 0.05.

**Figure 6 ijerph-19-12393-f006:**
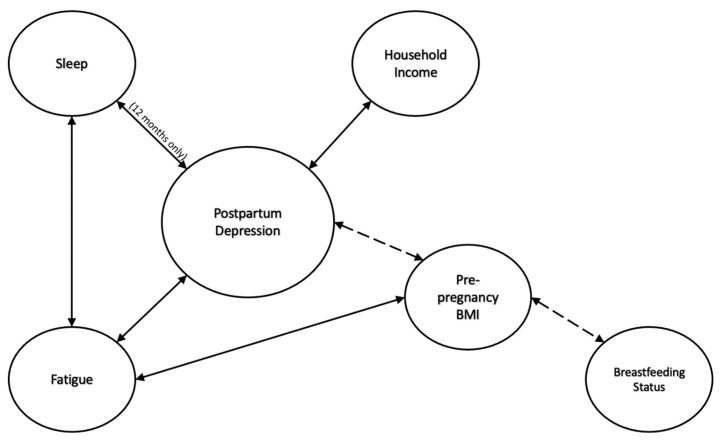
Summary of Study Findings.

**Table 1 ijerph-19-12393-t001:** Descriptive statistics for the participants.

	Pre-Pregnancy (n = 26)	6 months Postpartum (n = 26)	12 months Postpartum (n = 17)
**BMI**	24.5 ± 4.0	25.2 ± 4.4	24.2 ± 4.5
**BMI Classification**			
**Underweight < 18.5**	0 (0%)	0 (0%)	0 (0%)
**Normal Weight 18.5–24.9**	15 (63%)	15 (58%)	12 (70%)
**Overweight 25–29.9**	7 (29%)	7 (27%)	3 (18%)
**Obese ≥ 3**	2 (8%)	4 (15%)	2 (12%)
**Weight**	68.1 ± 11.7 kg	70.3 ± 13.2 kg	67.6± 13.6 kg
**Body Fat Percentage**		26.9 ± 6.9 %	23.8 ± 5.9%
**Age**	32.0 ± 4.3 years		
**Race** **White**	100%		
**Parity**			
**Nulliparous**	10 (38%)		
**Multiparous**	16 (62%)		
**Breastfeeding Status**			
**Breastfeeding**		15 (57%)	6 (35%)
**Formula-Feeding**		7 (27%)	5 (29%)
**Both/Combination**		3 (12%)	3 (18%)
**Did Not Report**		1 (4%)	3 (18%)
**Physical Activity Levels**			
**Sedentary (%)**		54.8 ± 9.4%	55.2 ± 15.8%
**Light (%)**		32.1 ± 7.3%	31.0 ± 10.5%
**Moderate (%)**		12.9 ± 3.9%	13.5 ± 7.09
**Edinburg Postnatal Depression Scale Scores**			
**Median**		4.0	3.0
**Interquartile Range**		6.5	5.5
**Household Income**	Average: USD 99,583 Range: USD 28,000–USD 240,000		

**Table 2 ijerph-19-12393-t002:** Dietary intake and postpartum depression.

	Relationship between Dietary Variable and Postpartum Depression Scores (EPDS)	
	6 months	12 months
**Total kilocalories**	r = 0.010, *p* = 0.965	r = −0.150, *p* = 0.609
**Vitamin B6**	r = 0.100, *p* = 0.657	r = −0.116, *p* = 0.693
**Zinc**	r = 0.051, *p* = 0.818	r = −0.020, *p* = 0.945
**Selenium**	r = −0.089, *p* = 0.695	r = 0.050, *p* = 0.925

**Table 3 ijerph-19-12393-t003:** Simple and Multiple Linear Regression.

Simple Linear Regression					
Predictors	B	SE	β	*p*-value	95% CI
Pre-pregnancy BMI *	28.670	11.542	0.468	0.021 *	4.733–52.607
Household Income	−6.631	3.406	−0.391	0.065	−13.714–0.452
Fatigue *	13.686	3.564	0.642	<0.001 *	6.274–21.099
Breastfeeding Status *	3.630	1.736	0.407	0.048 *	0.030–7.230
**Multiple Linear Regression**					
Predictors	B	SE	β	*p*-value	95% CI
Pre-pregnancy BMI	4.269	12.269	0.072	0.732	−21.615–30.153
Household Income *	−6.405	2.827	−0.375	0.037 *	−12.370–−0.440
Fatigue *	10.854	3.562	0.524	0.007 *	3.338–18.369
Breastfeeding Status	1.692	1.760	0.193	0.350	−2.022–5.406

* *p* < 0.05. B, unstandardized coefficients; SE, coefficients standard error; β, standardized coefficients; R-squared: 0.593; Adjusted R-squared: 0.497; DOF: 21, *p* = 0.003.

## Data Availability

The data presented in this study are available on request from the corresponding author.

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
