# Peer review of "Modifiable Maternal Factors and Their Relationship to Postpartum Depression"

_ijerph, 2022, doi:10.3390/ijerph191912393_

Round 1
Reviewer 1 Report
In the manuscript authors present the useful information about postpartum depression. Please clarify how did the authors obtain informed consent from participants of this Survey. Is the sample of 26 respondents large enough? Which test was used to test dana for normality 'distribution?Discussion:
Please discuss the limitations of the study. Discuss the results of this study and
compare them with existing knowledge in the literature.
Literature: Newer literature should be used. Check the Instructions for the author list
references according to the journal's recommendations
Key words: MeSH indexed key words should be used
Author Response
|
Reviewer #1 |
|
|
Please clarify how did the authors obtain informed consent from participants of this Survey. |
Details on the process are outlined in the previously published paper “Postpartum Metabolism: How Does It Change from Pregnancy and What are the Potential Implications?”. In summary, women were consented for the study visit and to take the surveys while face-to-face in the laboratory. The consent form was verbally explained and signatures were obtained (including a witness).
A line about this was added to the manuscript.
“…the participants were provided with a written informed consent form, which was verbally explained prior to obtaining a signature while present in the lab.” |
|
Is the sample of 26 respondents large enough? |
We acknowledge that this is a potential limitation, and for this reason, there were certain statistical analyses that we were unable to conduct (i.e. regression with 12 month data). However, our sample size was based on the number of women in the larger trial who agreed to come back for return visits. Some of them had relocated and others became less interested in participating in 5 hour study visits when they had an infant to care for. Therefore, our sample size was not something we were able to increase despite our best efforts; however, we certainly acknowledge this as a limitation. That in mind, we were excited to still find significance among many outcomes studied, which likely suggests we would continue to see these relationships, albeit likely stronger, with an increased sample size and increased power.
Small sample size is acknowledged in the discussion. |
|
Which test was used to test dana for normality 'distribution? |
Shapiro-Wilk tests were used (this is included in the methods). |
|
In this discussion: |
|
|
Please discuss the limitations of the study. |
The limitations of the study are outlined in the discussion section. The three primary limitations are the small sample size, lack of mental health data from prior to delivery, and the lack of racial diversity of the participant population. |
|
Discuss the results of this study and compare them with existing knowledge in the literature. |
Thank you for this comment. We systematically went through each of our reported outcomes to make sure we discussed them in the context of existing work.
We added several items to the discussion with several sources from the last 3-5 years.
There is evidence of a bidirectional association between maternal mental health status and breastfeeding35. Interventions offering individualized support (from professionals and peers in a variety of settings) have been successful at improving both mental health and breastfeeding outcomes35.
The American College of Obstetricians and Gynecologists recommends that healthcare providers screen mothers at least once for depression and anxiety symptoms during the perinatal period using a standardized and validated tool and that they complete a full assessment of mood and emotional well-being during the comprehensive postpartum visit19. Several risk factors for PPD have been previously identified, but the true cause is not yet to be understood and likely differs per individual. Previous research suggests that the cause of PPD is multifactorial, likely including a combination of mental, socioeconomic, physical, and psychosocial factors20.
|
|
Literature: Newer literature should be used. Check the Instructions for the author list references according to the journal's recommendations |
We added several sources from the last 3-5 years to the discussion section. We double checked all of the instructions for references to ensure we are compliant with the reference guidelines. As such, we made two key formatting changes throughout all of the citations. Thank you for noticing this!
We also changed a few sentences to people first language and left those highlighted for transparency. |
Reviewer 2 Report
This is a very well-designed study to identify factors that are associated with postpartum depression (PPD) that can be modified; it was carried out with great attention to details and with a good number of survey measures. Also notable is that these many factors were evaluated at the same time, instead of studies that evaluate only one element and its relationship to PPD. For these things the authors are to be commended.
Page 9, the paragraph about the relationship between BMI and PPD: Reference 19 (Silverman) shows that during first trimester, extremes of BMI in either direction (low or high) were associated with increased risk of PPD.
The authors have called household income a “modifiable” factor. On a large scale, over time, this is certainly true, and during and after pregnancy perhaps it could be with government-provided income supplements or something of the like. But realistically the usual way that income would be modified during the postpartum period is in a downward direction due to new mothers not continuing to work if they stay home to care for their infants. So some of the factors which they have called “modifiable” are amenable to change in the short-term (nutrition, sleep, breastfeeding), but some are much less so. I don’t know if it makes sense for the authors to comment on this or not.
In their discussion around fatigue, the authors discuss how maternal fatigue could contribute to the occurrence of PPD. However, they have left virtually unexamined and unstated the fact that fatigue can be a presenting symptom of PPD. Again, I believe this deserves mention. Which comes first, the fatigue or the PPD? This cannot be answered.
The authors tried to identify the shortcomings of their study, but they left one major shortcoming out: the lack of racial diversity in their patient population. All of their participants were white. They do not state what proportion of the original study group of 64 women were white vs. minorities. They do not adequately describe the demographics of the overall patient population from which they drew their study participants. This lack of racial diversity should at least be acknowledged, as research does show that many sociodemographic factors adversely impact racial minorities compared with whites, and these in turn could affect rates of PPD. And, African-American and Hispanic mothers are at higher risk for reporting early postpartum depressive symptoms compared with white mothers (Howell et al, 2005, Obstet Gynecol). When summarizing their findings, they should note that they are applicable to white middle and upper class women, not to all women.
In summary, this is a valuable—if small—study that needs only minor modifications before acceptance for publication.
Author Response
|
Reviewer #2 |
|
|
Page 9, the paragraph about the relationship between BMI and PPD: Reference 19 (Silverman) shows that during first trimester, extremes of BMI in either direction (low or high) were associated with increased risk of PPD. |
This is a great point. We failed to acknowledge that low BMI could also be a factor. We did not have anyone that was on the extreme low end so did not think to mention this. Our relationship was linear and not U-shape, but it makes sense that this has been seen in other studies.
We added a sentence to the discussion on this.
Previous work has described a relationship between weight status and PPD. Silverman et al. reported extremes in BMI (in either direction) are associated with an increased risk of PPD21. |
|
The authors have called household income a “modifiable” factor. On a large scale, over time, this is certainly true, and during and after pregnancy perhaps it could be with government-provided income supplements or something of the like. But realistically the usual way that income would be modified during the postpartum period is in a downward direction due to new mothers not continuing to work if they stay home to care for their infants. So some of the factors which they have called “modifiable” are amenable to change in the short-term (nutrition, sleep, breastfeeding), but some are much less so. I don’t know if it makes sense for the authors to comment on this or not. |
This is a great perspective. We failed to elaborate on the degree to which some factors could be modified in relation to others. While income may be very difficult to modify, it is still modifiable as opposed to nonmodifiable factors like race, age, and genetics.
We added a sentence to the discussion to address this:
While each of the maternal factors analyzed in the present study are potentially modifiable, it should be acknowledged that some can be modified more easily than others. Income, specifically, is a factor that women in certain life circumstances may not have the means to change. It is certainly possible that some mothers may experience a drop in pay during the postpartum period if unable to work for a period of time with a job that does not have maternity leave policies in place. Some women may also opt for extended unpaid leave which may impact income.
|
|
In their discussion around fatigue, the authors discuss how maternal fatigue could contribute to the occurrence of PPD. However, they have left virtually unexamined and unstated the fact that fatigue can be a presenting symptom of PPD. Again, I believe this deserves mention. Which comes first, the fatigue or the PPD? This cannot be answered. |
The sentence below was added to address this comment.
Fatigue is one of the presenting symptoms of depression and is part of the diagnostic criteria for depressive disorders29. The question of whether fatigue causes PPD or if PPD causes fatigue has yet to be answered, and likely a vicious cycle, but the findings presented in the present study support the existing idea that PPD and fatigue are correlated. |
|
The authors tried to identify the shortcomings of their study, but they left one major shortcoming out: the lack of racial diversity in their patient population. All of their participants were white. They do not state what proportion of the original study group of 64 women were white vs. minorities. They do not adequately describe the demographics of the overall patient population from which they drew their study participants. This lack of racial diversity should at least be acknowledged, as research does show that many sociodemographic factors adversely impact racial minorities compared with whites, and these in turn could affect rates of PPD. And, African-American and Hispanic mothers are at higher risk for reporting early postpartum depressive symptoms compared with white mothers (Howell et al, 2005, Obstet Gynecol). When summarizing their findings, they should note that they are applicable to white middle and upper class women, not to all women. |
We added a paragraph addressing the lack of racial diversity in the participant population to the discussion.
An additional limitation of the study is the lack of racial diversity in the participant population. All the participants were white. According to the 2021 U.S. census, 72.5% of the population in Bowling Green, Kentucky is white39. Thus, the study sample is not a direct representation of the racial breakdown of pregnant women in Bowling Green, Kentucky. Of the 64 women recruited for the original study: 62 were white, one was Hispanic, and one was African American. Previous research has shown that many sociodemographic factors negatively impact racial minorities40. The negative impact of these factors could affect PPD scores. African American and Hispanic mothers have a higher risk of reporting early PPD symptoms in comparison to white counterparts41. It should be noted that the findings of this study are only applicable to women who are white and middle or upper-class. Future studies should make an effort to include underrepresented groups who are more likely to experience poor pregnancy outcomes including PPD.
|